# Copper Cobalt Sulfide Structures Derived from MOF Precursors with Enhanced Electrochemical Glucose Sensing Properties

**DOI:** 10.3390/nano12091394

**Published:** 2022-04-19

**Authors:** Daojun Zhang, Xiaobei Zhang, Yingping Bu, Jingchao Zhang, Renchun Zhang

**Affiliations:** 1College of Chemistry and Chemical Engineering, Anyang Normal University, Anyang 455000, China; zzuczxb@126.com (X.Z.); m18337278507@163.com (Y.B.); zjc19830618@126.com (J.Z.); rczhang@aynu.edu.cn (R.Z.); 2College of Chemistry, Zhengzhou University, 100 Science Road, Zhengzhou 450001, China

**Keywords:** copper cobalt sulfide, porous structures, nonenzymatic glucose sensing, electrocatalysts

## Abstract

Nonenzymatic electrochemical detection of glucose is popular because of its low price, simple operation, high sensitivity, and good reproducibility. Co-Cu MOFs precursors were synthesized via the solvothermal way at first, and a series of porous spindle-like Cu-Co sulfide microparticles were obtained by secondary solvothermal sulfurization, which maintained the morphology of the MOFs precursors. Electrochemical studies exhibit that the as-synthesized Cu-Co sulfides own excellent nonenzymatic glucose detection performances. Compared with CuS, Co (II) ion-doped CuS can improve the conductivity and electrocatalytic activity of the materials. At a potential of 0.55 V, the as-prepared Co-CuS-2 modified electrode exhibits distinguished performance for glucose detection with wide linear ranges of 0.001–3.66 mM and high sensitivity of 1475.97 µA·mM^−1^·cm^−2^, which was much higher than that of CuS- and Co-CuS-1-modified electrodes. The constructed sulfide sensors derived from MOF precursors exhibit a low detection limit and excellent anti-interference ability for glucose detection.

## 1. Introduction

Currently, diabetes as a common chronic disease is already a serious threat to human health. Therefore, developing a simple and sensitive detection method for glucose is important for clinical diagnosis and diabetes management [1,2]. Compared with colorimetry, spectroscopy, and fluorescence analytical methods, nonenzymatic electrochemical glucose detection has received widespread attention due to its low cost, simple operation, and high sensitivity [3,4,5]. In recent years, transition-metal oxides (TMOs) and transition-metal sulfides (TMSs) have been exploited as advanced electrocatalysts to construct high performance electrochemical sensors [6,7,8,9,10,11,12,13,14,15,16,17,18,19].

Recently, among the transition-metal based electrodes materials, copper-based oxides and sulfides with various morphologies and structures have been used as electrode materials for nonenzymatic electrochemical glucose detection. For instance, Cu/Cu_2_O hollow microspheres were prepared by solvothermal conditions and exhibited a high catalytic activity for glucose oxidation [20]. The glucose electrochemical sensor constructed by CuO nanorod dispersed hollow carbon fibers (CuO NR @ PCFs) [21] showed a wide linear range (0.005–0.8 mM, 0.8–8.5 mM) and a low detection limit (0.1 μM). The reported flower-like CuCo_2_O_4_/C microspheres [22] -constructed sensor exhibited a wide linear range and low detection limit. The Cu_x_Co_3-x_O_4_ nano-needle framework thin-film electrode reported by Xu [23] exhibited an ultrahigh sensitivity of 13,291.7 µA·mM^−1^·cm^−2^ for glucose detection. CuS nanotubes were prepared in an O/W microemulsion system at low temperature [24], and the glucose concentration could be detected by the CuS nanotube sensor with high sensitivity (7.842 µA·µM^−1^). Karikalan et al. synthesized S-rGO/CuS nanocomposites to construct an electrochemical glucose sensor, and the linear concentration range of the constructed sensor was 0.0001–3.88 mM and 3.88–20.17 mM, respectively, and the detection limit of 32 nM was quite low [25]. Xu et al. demonstrated the synthesis of CuCo_2_S_4_ nanosheets on flexible carbon fiber textiles (CFT) by a hydrothermal method [26]. The sensor constructed of CuCo_2_S_4_ nanosheets had a high sensitivity of 3852.7 µA·mM^−1^·cm^−2^ and a linear range up to 3.67 mM. Compared with related transition metal oxides, copper cobalt sulfides are more suitable as electrode materials for nonenzymatic glucose sensors due to the improved electrical conductivity [26].

In recent years, metal−organic frameworks (MOFs) have served as self-sacrificial precursors for preparation of porous micro-/nanostructured transition metal oxides and sulfides [27,28,29,30]. The MOF-derived TMOs and TMSs usually exhibit porous structures and high surface areas with enhanced electrocatalytic and electrochemical energy storage performances [31,32,33,34,35]. In this work, the shuttle-like copper cobalt sulfide structures were synthesized via MOF sacrificial templates. The electrochemical properties of copper cobalt sulfide -constructed electrodes were studied by cyclic voltammetry and the amperometric method. At a potential of 0.55 V, the linear range of Co-CuS-2 modified electrode was 0.001–3.66 mM with a detection limit of 0.1 μM, and the sensitivity of the electrode was 1475.97 µA·mM^−1^·cm^−2^. The results indicate that the sensor owns good electrochemical sensing performance for glucose and has a potential application in glucose detection.

## 2. Experimental Section

### 2.1. Chemicals 

Polyvinylpyrrolidone (PVP) and 2,5-dihydroxyterephthalic acid (H_4_dobdc) were purchased from Shanghai Macklin Biochemical Co., Ltd. (Shanghai, China); Cu(NO_3_)_2_·4H_2_O, Co(NO_3_)_2_·6H_2_O, glucose (Glu), ascorbic acid (AA), and NaOH were purchased from Sinopharm Chemical Reagent Co., Ltd. (Shanghai, China); Thioacetamide (TAA), uric acid (UA), dopamine (DA), sodium chloride (NaCl), glutathione (GSH), and sucrose (Suc) were purchased from Aladdin Industrial Corporation (Shanghai, China); Ethylene glycol, ethanol, and N,N-dimethylformamide (DMF) were purchased from Tianjin Fuyu Fine Chemical Co., Ltd. (Tianjin, China). All chemicals and solvents were used without further purification.

### 2.2. Preparation of Spindle-like Cu-Co Sulfide Microparticles

The solvent system of Cu-Co MOF precursors synthesis is similar to that of our previously reported paper [36]. For the preparation of Co-Cu MOF precursors, Cu(NO_3_)_2_·4H_2_O, Co(NO_3_)_2_·6H_2_O, 2,5-dihydroxyterephthalic acid (H_4_dobdc), and polyvinylpyrrolidone (PVP) were added in DMF/ethanol/water mixed solvent according to the molar ratio of Co/Cu of 8:2, 7:3 and 0:1, the mixture was heated at 100 °C for 12 h, and then MOF precursors were isolated and washed twice with DMF and water. Cu-Co sulfides were synthesized via an effective sulfurization treatment of MOF precursors with thioacetamide (TAA), the samples with Co/Cu ratio from high to low denoted as Co-CuS-1 and Co-CuS-2, respectively. The MOF precursors were redispersed into 3 mL of ethylene glycol. Then, 0.0043 g of TAA was added and fully stirred, the mixture was transferred into a 25 mL stainless-steel Teflon-lined autoclave and reacted at 110 °C for 12 h, and the black powder was obtained and washed three times with ethanol and water.

### 2.3. Materials Characterization

Powder X-ray diffraction (PXRD) analyses of the as-prepared samples were conducted on a PANalytical X’Pert PRO MPD system with Cu_K__a_ radiation (λ = 1.5418 Å) and operated at 40 kV and 40 mA. The morphologies and compositions were analyzed by scanning electron microscopy (SEM) with energy dispersive spectroscopy (EDS) on a Hitachi SU-8010 instrument and X-ray photo-electron spectroscopy (XPS) using Thermo-Scientific system. The specific surface areas of the samples were acquired by N_2_ adsorption/desorption isotherms measured on a Gemini VII 2390 analyzer at 77 K.

### 2.4. Electrode Preparation and Measurement

All electrochemical tests were conducted on a CHI660E electrochemical workstation with a typical three-electrode system. First, 2 mg Co-Cu sulfides was dispersed in 1.0 mL distilled H_2_O via ultrasound 30 min. A glassy carbon electrode (GCE) with diameter 3 mm was polished with alumina slurries and washed with ultrapure H_2_O. Afterward, 5 μL of the suspension was covered onto the GCE surface to obtain Co-CuS/GCE. The modified electrode was used as the working electrode, Ag/AgCl was used as the reference electrode, and Pt wire was used as the counter electrode.

## 3. Results and Discussion

The morphologies of Co-Cu MOF precursors were analyzed by scanning electron microscopy (SEM) technique and are shown in Figure 1, which exhibited a spindle-like structure with a well-distributed and smooth surface. Figure 2 shows the SEM images of derived samples of CuS (Figure 2a,b), Co-CuS-1 (Figure 2d) and Co-CuS-2 (Figure 2e). After effective sulfurization treatment of MOF precursors with TAA, the derived products can largely retain the morphology of MOF precursors; however, the surface of all samples seems rough and porous. According to the EDS mapping images of a single shuttle-like CuS particle (Figure 2c), the Cu and S elements are evenly distributed. Appendix A and Figure 2f show the EDS mapping images of Co-CuS-1 and Co-CuS-2, respectively. It can be seen from the images that there is a distribution of Co, Cu and S elements, which indicates that Co element is doped into CuS microparticles. The phase of as-synthesized sulfides was checked by the XRD patterns and are exhibited in Figure 3a; all the positions of the peaks are consistent with the standard card number JCPDS No.065-3588 of the hexagonal phase CuS, and no other impurity peaks appear in the patterns. It can be deduced that partial-doped cobalt ions into CuS do not change its crystal structure. Figure 3b shows the N_2_ adsorption isotherm of the Co-CuS-2 sample and the corresponding pore size distribution. Co-CuS-2 is the type IV adsorption isotherm, which belongs to the typical mesoporous structure. The specific surface area calculated by the BET method is 16.3 m^2^ g^−1^, and the average pore size is 31.72 nm. The element content of the Co-CuS-1 and Co-CuS-2 samples were further characterized by EDS, which indicated that the Co/Cu ratios are close to the stoichiometric ratio of raw materials (Figure 4a). XPS technique was further used to analyze the surface valance state of Co and Cu in the corresponding sulfide. Appendix A provides the XPS survey spectra of the as-synthesized samples. The high revolution spectra are shown in Figure 4b–d. The high revolution spectra of Cu in the three samples are similar, the binding energy of the two peaks located at 931.4 and 951.3 eV was attributed to Cu 2p_3/2_ and Cu 2p_1/2_, respectively [25]. The fitted peaks of 931.4 and 951.2 eV indicate the existence of Cu^+^, and the peaks at 932.4 and 953.1 eV correspond to Cu^2+^. The high resolution Co 2p spectra in Co-CuS-1 are fitted with two doublet peaks centered at 780.5 and 796.4 eV, which correspond to Co^3+^. The peaks at 781.9 and 797.5 eV correspond to Co^2+^. Figure 4d shows the S 2p spectra, with the peaks centered at 161.2 and 162.4 eV for S 2p_3/2_ and S 2p_1/2_, respectively. For Co-CuS-2, the peak of 163.6 eV increased dramatically, which can be ascribed to a Metal-S bond at a low coordination environment and contributed to an increase in the intrinsic conductivity [37,38].

Figure 5a shows the CV curves of the bare electrode and copper cobalt sulfide-modified electrodes in the 0.1 M NaOH electrolyte containing 1 mM Glu. As shown in Figure 5a, the bare electrode has almost no response, CuS- and Co-CuS-1-modified electrodes have weak redox peaks, and the Co-CuS-2-modified electrode has a pair of obvious redox peaks at 0.45/0.60 V (vs. Ag/AgCl), indicating that the Co-CuS-2 sample exhibits the best response to glucose. As shown in Figure 5b–d, the CV curves of CuS, Co-CuS-1, and Co-CuS-2 at different glucose concentrations further show that the redox peaks of the Co-CuS-2 electrode is the strongest, indicating that this material has the best electrocatalytic performance for glucose among the three samples. The possible oxidation mechanism of glucose may be described in the following three steps [22,23,26]:CuCo-S +OH^−^ + H_2_O → CuSOH + CoSOH + e^−^(1)CoSOH +OH^−^ → CoSO + H_2_O + e^−^(2)CuOSH + CoSO + glucose → CuS + CoSOH + glucolactone(3)

CuSOH and CoSO intermediator might be formed through electrooxidation at alkaline conditions. The formed CuSOH and CoSO adsorbed glucose molecules and subsequently oxidated to gluconolactone in an alkaline medium.

In order to further acquire the kinetic information of glucose electrocatalytic oxidation of the CuS, Co-CuS-1, and Co-CuS-2 electrodes, the CV curves of three samples at scan rates varying from 20 to 180 mV s^−1^ were studied. Figure 6a,c,e shows the CV curves of CuS, Co-CuS-1, and Co-CuS-2 in 0.1 M NaOH solution containing 1 mM glucose at different scan rates. The peak current of the three samples increases steadily with the increase in scan rate. As seen in Figure 6b,d, the peak current (anodic and cathodic) of the CuS and Co-CuS-1 electrodes increases linearly with the scan rate, and the fitted linear equations are *I*_pa_ = 0.626*v* + 31.172, *I*_pc_ = −0.489*v* + 12.852, and *I*_pa_ = 0.634*v* + 28.708, *I*_pc_ = −0.717*v* +17.782 respectively, indicating that CuS and Co-CuS-1 electrodes are adsorption-controlled processes for oxidation glucose. Figure 6f shows that both the anodic and cathodic peak current has a linear relationship with the square root of the scan rate, indicating that Co-CuS-2 is a diffusion-controlled process. The fitted linear equations are *I*_pa_ = 21.176*v*^1/2^ + 47.474, *I*_pc_ = −15.716*v*^1/2^ + 8.971, respectively, and this may be attributed to the large surface area and good conductivity; thus, it is conducive to glucose detection.

In order to systematically study the effect of working potential on the electrocatalytic oxidation of glucose for CuS-, Co-CuS-1-, and Co-CuS-2-modified electrodes, the current−time (*I*–t) curves at different potentials were measured. Figure 7a shows the current response (0.1 M NaOH) of the CuS electrode via a continuously increasing glucose concentration at 0.5, 0.55 and 0.6 V. The CuS electrode exhibits the highest amperometric response at 0.6 V. The corresponding calibration curve at 0.6 V is also shown in Figure 7b. The linear range of the CuS electrode is 0.002–2.16 mM, the sensitivity is 905.42 µA·mM^−1^·cm^−2^, and the limit of detection (LOD) of 0.9 μM is calculated based on 3σ/s, where σ is the standard deviation of the blank, and s is the slope of the calibration curve [12,39].

Figure 8a,b shows the *I*–t curves of Co-CuS-1 and Co-CuS-2 at different potentials, which shows the best glucose electrocatalytic performance at a potential of 0.55 V after Co doping. Thus, the electrocatalytic properties of CuS, Co-CuS-1 and Co-CuS-2 at 0.55 V were compared and are shown in Figure 8c. It can be seen from the image that the amperometric response of Co-CuS-2 is the highest. Figure 8d shows the calibration curve of CuS at 0.55 V; the linear range is 0.002–2.66 mM, the sensitivity is 686.13 µA·mM^−1^·cm^−2^, and the detection limit is 0.4 μM (3σ/s). Figure 8e shows the calibration curve of Co-CuS-1; the linear range is 0.001–3.16 mM, the sensitivity increases to 1206.75 µA·mM^−1^·cm^−2^, and the detection limit decreases to 0.3 μM (3σ/s). Figure 8f shows the calibration curve of Co-CuS-2 at 0.55 V; the linear range extends from 0.001 mM to 3.66 mM, the sensitivity is 1475.97 µA·mM^−1^·cm^−2^, and the detection limit is 0.1 μM (3σ/s). Table 1 shows the details. As compared, at a potential of 0.55 V, Co-CuS-2 has the widest linear range, the highest sensitivity and lowest LOD among the three electrodes. The sensitivity of the Co-CuS-2 sensor is higher than that of the Cu/Cu_2_O hollow microspheres [20], CuCo_2_O_4_/C microspheres [22], hierarchical Co_3_O_4_ film [39], CuO_x_-CoO_x_/graphene [40], Octahedral Cu_2_O [41], and CuO microspheres [42]; however, it is lower than that of the CuCo_2_S_4_ nanosheets [26] and NiCo_2_O_4_ hollow nanorods [43]. The comprehensive performance of Co-CuS-2/GCE is equivalent to or better than that of previously reported electrochemical glucose sensors (Table 2).

As shown in Figure 9a, all Nyquist diagrams contain the semicircular part at high frequency and the oblique line at low frequency. The tilt line is related to the diffusion limit step, and the *R*_ct_ of the electrode surface can be equal to the radius of the semicircular part. The radius of Co-CuS-2 is the smallest, indicating that the conductivity of Co-CuS-2 is the largest, which is one of the reasons for its best electrocatalytic performance for glucose. The repeatability of CuS, Co-CuS-1 and Co-CuS-2 electrodes were tested and are shown in Figure 9b. The current response of adding 200 μM Glu 13 times in 0.1 M NaOH solution at a potential of 0.55 V shows that the current response of Co-CuS-2 is high, the step change is almost unchanged, and the calculated RSD is 4.19%. In order to evaluate the selectivity of the constructed electrodes to glucose detection, as shown in Figure 9c, three different sulfide electrodes showed good anti-interference performance at a working potential of 0.55 V. The specific operation is to add 200 μM glucose (Glu), 20 μM ascorbic acid (AA), uric acid (UA), dopamine (DA), sodium chloride (NaCl), glutathione (GSH), sucrose (Suc) and 200 μM glucose (Glu) in 0.1 M NaOH supporting solution. It can be seen from the *I*–t curves that the response current for glucose of the Co-CuS-2-modified electrode remains unchanged after adding interfering substances, while the response current of interferents is almost negligible, indicating that the electrode has good selectivity for glucose detection. Figure 9d shows the stability of the as-synthesized sample-modified electrodes. After adding 200 μM Glu solution, the current response lasts for 3500 s. The results show that the retention rates of Co-CuS-2, Co-CuS-1 and CuS are 93%, 76% and 84%, respectively, indicating that Co-CuS-2 has the best stability. Therefore, the Co-CuS-2-modified electrode has good reproducibility, selectivity, and stability for the detection of glucose.

## 4. Conclusions

Co/Cu MOFs precursors were synthesized by the one-step solvothermal method, and a series of porous spindle-like Cu-Co sulfide microparticles were obtained by secondary solvothermal sulfurization, which maintained the morphology of the MOFs precursors. The porous structure of these materials was conducive to the diffusion of electrolytes and analytes. Compared with CuS, Co (II) ion doping can improve the conductivity and electrocatalytic activity of the materials. At a potential of 0.55 V, the linear range of the Co-CuS-2 electrode for glucose detection was 0.001–3.66 mM with an LOD of 0.1 μM, and the sensitivity was 1475.97 µA·mM^−1^·cm^−2^, which was much better than that of the CuS and Co-CuS-1 samples. This work provides an effective strategy for glucose detection and opens up a new way to improve the electrochemical performance of non-enzyme electrochemical sensors.

## Figures and Tables

**Figure 1 nanomaterials-12-01394-f001:**
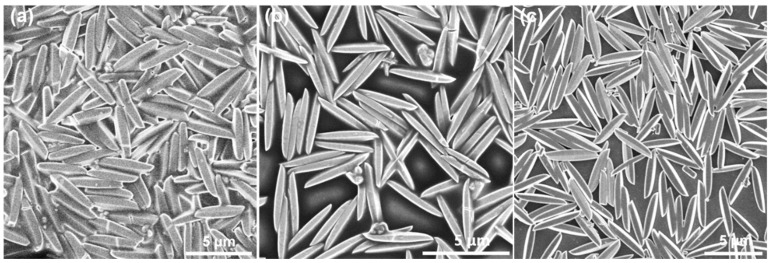
SEM images of (**a**) Cu-MOF, (**b**) CuCo-MOF-1, and (**c**) CuCo-MOF-2 precursors.

**Figure 2 nanomaterials-12-01394-f002:**
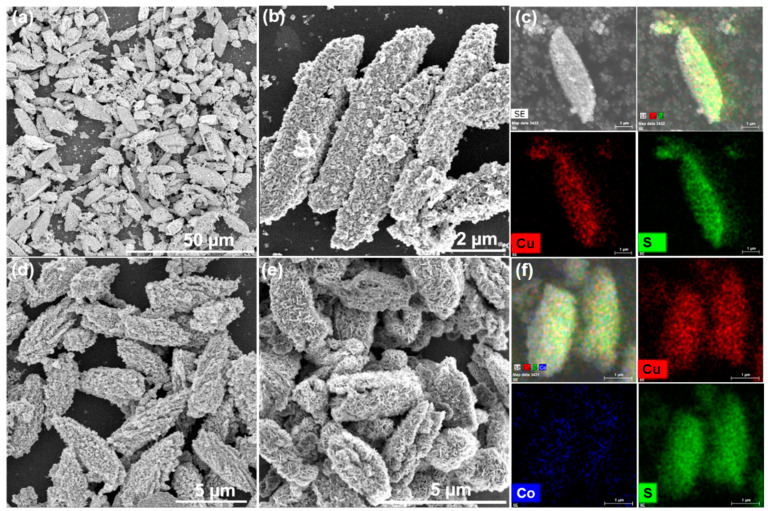
SEM images of copper cobalt sulfides (**a**,**b**) CuS, (**d**) Co-CuS-1, (**e**) Co-CuS-2, and the EDX mapping element distribution of (**c**) CuS, (**f**) Co-CuS-2.

**Figure 3 nanomaterials-12-01394-f003:**
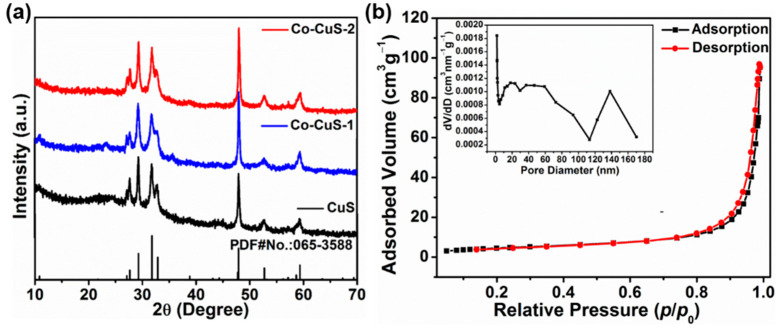
(**a**) XRD patterns of the samples with different proportions; (**b**) N_2_ adsorption isotherm and pore distribution of Co-CuS-2.

**Figure 4 nanomaterials-12-01394-f004:**
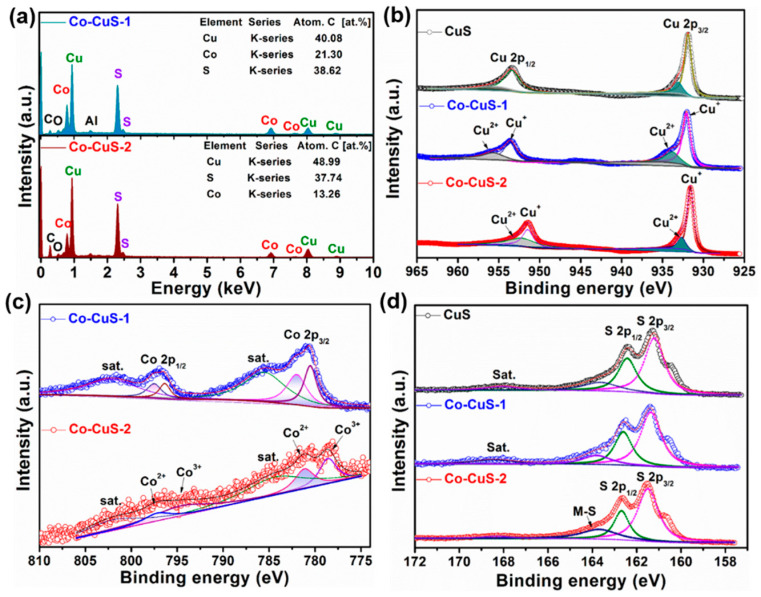
(**a**) EDS of CuS-1 and Co-CuS-2 samples. High resolution XPS spectra of Cu (**b**), Co (**c**), and S (**d**) for CuS, Co-CuS-1, and Co-CuS-2 samples.

**Figure 5 nanomaterials-12-01394-f005:**
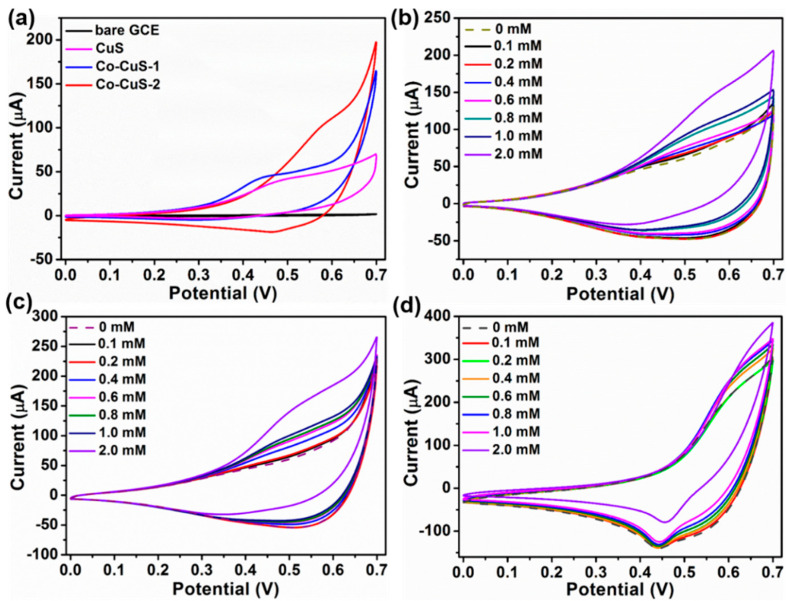
(**a**) CV curves of CuS, Co-CuS-1, Co-CuS-2 and bare electrode in 0.1 M NaOH electrolyte containing 1 mM Glu (20 mV·s^−1^). CV curves in NaOH solution with different glucose concentration: (**b**) CuS, (**c**) Co-CuS-1, and (**d**) Co-CuS-2 (100 mV·s^−1^).

**Figure 6 nanomaterials-12-01394-f006:**
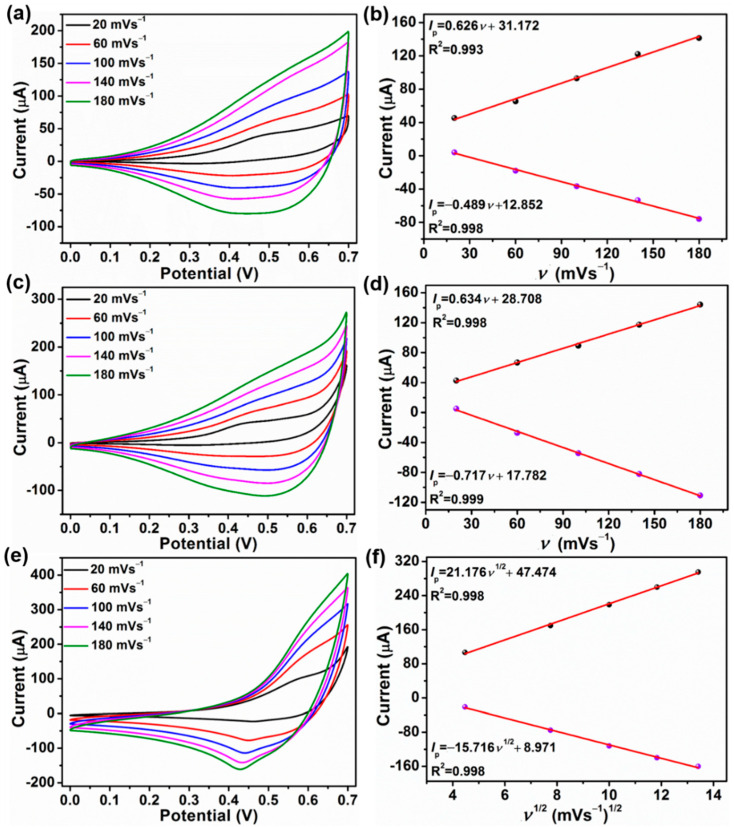
CV curves at different scan rates (20–180 mV s^−^^1^) in a 1 mM glucose solution (**a**) CuS, (**c**) Co-CuS-1, (**e**) Co-CuS-2, and the corresponding linear calibration curves, (**b**) CuS, (**d**) Co-CuS-1, (**f**) Co-CuS-2, respectively.

**Figure 7 nanomaterials-12-01394-f007:**
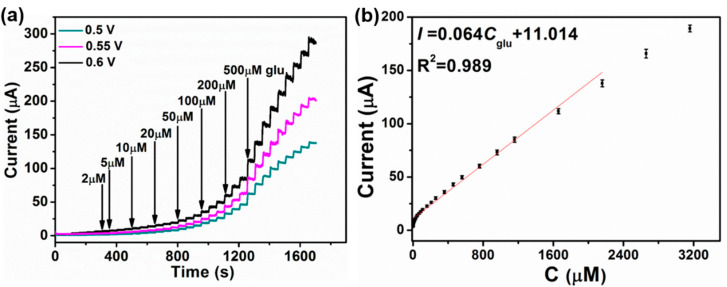
(**a**) Current−time curves of the CuS electrode with the continuous addition of glucose solution at 0.5, 0.55 and 0.6 V (0.1 M NaOH), respectively. (**b**) The calibration curves fitted from the current responses at 0.6 V.

**Figure 8 nanomaterials-12-01394-f008:**
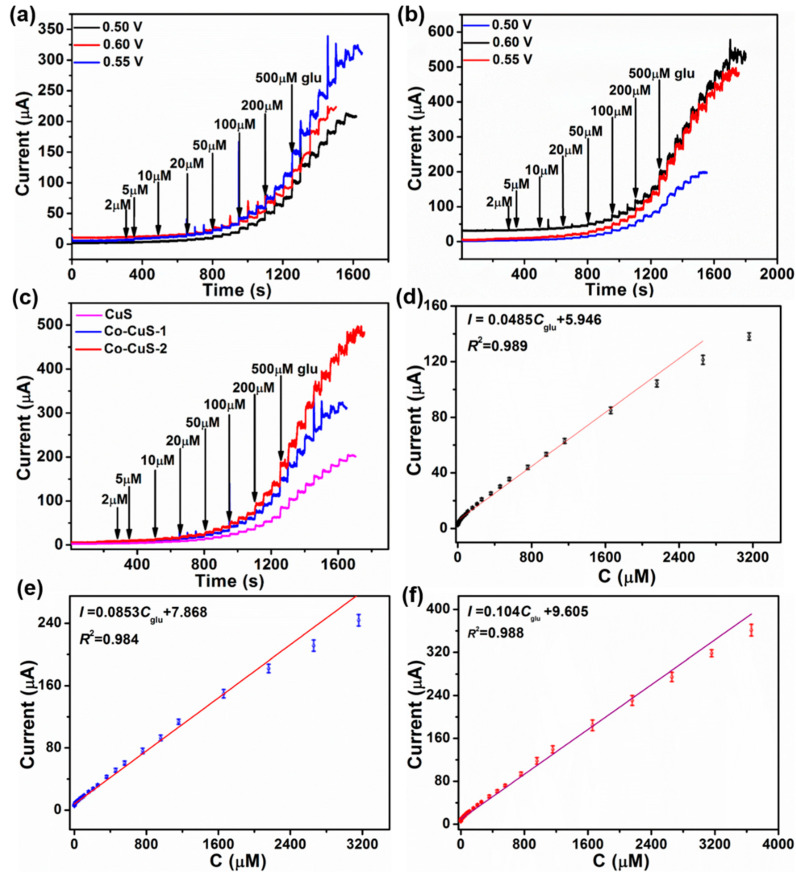
Current response of continuously increasing glucose concentration at 0.5, 0.55 and 0.60 V (0.1 M NaOH): (**a**) Co-CuS-1, (**b**) Co-CuS-2, (**c**) current response of CuS, Co-CuS-1 and Co-CuS-2 at 0.55 V. The corresponding calibration curves at 0.55 V: (**d**) CuS, (**e**) Co-CuS-1, (**f**) Co-CuS-2.

**Figure 9 nanomaterials-12-01394-f009:**
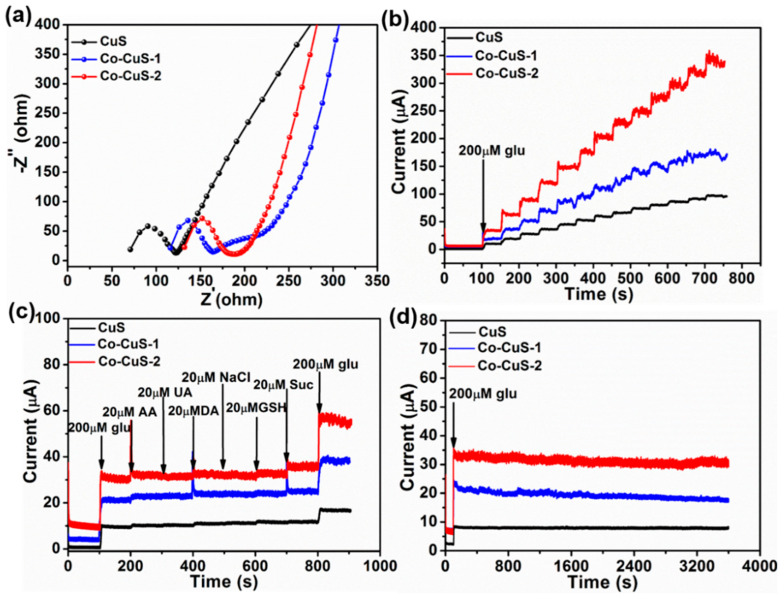
(**a**) Impedance diagram of CuS, Co-CuS-1 and Co-CuS-2 samples, (**b**) reproducibility, (**c**) anti-interference, and (**d**) stability.

**Table 1 nanomaterials-12-01394-t001:** Electrochemical sensing properties of Cu-Co sulfides with different doping ratios.

Electrode Material	Potential (V)	Linear Range (mM)	Detection Limit (µM)	Sensitivity (µA·mM^−1^·cm^−2^)
CuS	0.55	0.002–2.66	0.4	686.13
	0.60	0.002–2.16	0.9	905.42
Co-CuS-1	0.55	0.001–3.16	0.3	1206.75
Co-CuS-2	0.55	0.001–3.66	0.1	1475.97

**Table 2 nanomaterials-12-01394-t002:** Comparison of the Co-CuS-2 electrode with some reported sensors for glucose detection.

Electrode Material	Potential (V)	Linear Range (mM)	Detection Limit (μM)	Sensitivity (µA·mM^−1^·cm^−2^)	Ref.
Co-CuS-2	0.55	0.001–3.66	0.1	1475.97	This work
Cu/Cu_2_O hollow microspheres	0.45	0.22–10.89	0.05	33.63 µA·mM^−1^	20
CuO NR @ PCFs	0.60	0.005–0.8 0.8–8.5	0.1	608	21
CuCo_2_O_4/_C microspheres	0.60	0.005–8	1.5	707.71	22
CuS nanotube	0.20	0.05–5	–	7.842 µA·mM^−1^	24
CuCo_2_S_4_/carbon fiber textile	0.35	up to 3.67	1.01	3852.7	26
Co_3_O_4_ porous film	0.6	up to 3.0	1	366.03	39
CuO_x_-CoO_x_/ graphene	0.50 (vs. SCE)	0.005–0.57	0.5	507	40
Octahedral Cu_2_O	0.60	0.3–4.1	128	241	41
CuO microspheres	0.45 (vs. SCE)	0.001–4	0.5	349.6	42
NiCo_2_O_4_ hollow nanorods	0.60	0.0003–1	0.16	1685.1	43

## Data Availability

Not applicable.

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
