# Peer review of "Copper Cobalt Sulfide Structures Derived from MOF Precursors with Enhanced Electrochemical Glucose Sensing Properties"

_nanomaterials, 2022, doi:10.3390/nano12091394_

Round 1

Reviewer 1 Report

Authors employ CoCu sulfide derived from MOF for the detection of glucose. The work appears sound, however, there are some issues that need to be resolved before accepting the manuscript. They are listed below:

  • authors use abbreviations without explaining what they mean
  • analysis of sensing material has to be expanded. It would be nice to see the stoichiometry of metals and valance state of Co and Cu in the corresponding sulfide...
  • how was LoD calculated? It seems very low in comparison with the art.
  • authors need to explain the redox processes during the voltammetric scan, especially reduction, and why such big hysteresis?

Author Response

1 Comments: Authors employ CoCu sulfide derived from MOF for the detection of glucose. The work appears sound, however, there are some issues that need to be resolved before accepting the manuscript. They are listed below:

Comments: authors use abbreviations without explaining what they mean.

Response: We have explained the meaning of abbreviations in the revised manuscript.

Comments: analysis of sensing material has to be expanded. It would be nice to see the stoichiometry of metals and valance state of Co and Cu in the corresponding sulfide.

Response: The analysis of as-synthesized materials has to be expanded in this revised manuscript. We added EDS and XPS tests to analyze the stoichiometry of metals and valance state of Co and Cu in the corresponding sulfides. The element content of Co-CuS-1 and Co-CuS-2 samples were characterized by EDS, which indicated the Co/Cu ratios are close to the stoichiometric ratio of raw materials (Fig.4a). XPS technique was further used to analyze the surface valance state of Co and Cu in the corresponding sulfide. The high revolution spectra are shown in Figure 4b-d. The high revolution spectra of Cu in the three samples are similar, the binding energy of two peaks located at 931.4 and 951.3 eV is attributed to Cu 2p3/2 and Cu 2p1/2, respectively. The fitted peaks of 931.4 and 951.2 eV indicate the existence of Cu+, and the peaks at 932.4 and 953.1 eV corresponded to Cu2+. The high resolution Co 2p spectrum in Co-CuS-1 was fitted with two doublet peaks centered at 780.5 and 796.4 eV, which corresponds to Co3+. The peaks at 781.9 and 797.5 eV corresponded to Co2+. Figure 4d shows the S 2p spectra, the peaks centered at 161.2 and 162.4 eV for S 2p3/2 and S 2p1/2, respectively. For Co-CuS-2, the peak of 163.6 eV increased dramatically, which can be ascribed to metal-S bond at low coordination environment and contributed to increase the intrinsic conductivity.

Comments: how was LoD calculated? It seems very low in comparison with the art.

Response: In this manuscript, limit of detection (LOD) of the constructed sensors was calculated based on 3σ/s, where σ is the standard deviation of the background current and s is the slope of the calibration curves.

Comments: authors need to explain the redox processes during the voltammetric scan, especially reduction, and why such big hysteresis?

Response: In the revised manuscript, we explain the redox processes during the voltammetric scan. Redox peaks at 0.45/0.60 V (vs. Ag/AgCl) for Co-CuS-2 were observed on the CVs, which may be attributed to the redox reaction of Cu-CoS in the alkaline electrolyte

CuCo-S +OH- + H2O → CuSOH + CoSOH + e- (1)

CoSOH +OH- → CoSO + H2O + e- (2)

The big hysteresis in CV at high voltage is attributed to polarized and present oxygen evolution reaction.

Reviewer 2 Report

1-When the authors cite Randles-Sevcik equation, please check the value number it would be 2.69 105, please also add the definition of each parameters and their units, as ex.  D=7.6 10-6 cm2 s-1 for FeIII(CN)63-, etc..

2-in Fig. 5 b inset and in Figs. 6 d, e, f please add the error bars for all the data in the calibration plots and add in the text the replications of each mesurements.

Author Response

Comments: 1-When the authors cite Randles-Sevcik equation, please check the value number it would be 2.69 105, please also add the definition of each parameters and their units, as ex. D=7.6 10-6cm2s-1for FeIII(CN)63-, etc.

Response: Randles-Sevcik equation is not suitable for our constructed sensors, we deleted Randles-Sevcik equation in this revised manuscript.

2-in Fig. 5 b inset and in Figs. 6 d, e, f please add the error bars for all the data in the calibration plots and add in the text the replications of each mesurements.

Response: In the revised manuscript, we added the error bars for all the data in the calibration plots for Fig. 7 b inset and in Figs. 8 d, e, f in the revised manuscript.

Reviewer 3 Report

The author developed a highly sensitive electrochemical glucose sensor by using a nanoporous cobalt doped copper sulfide based electrode. In particular, they have succeeded in producing a highly sensitive non-enzymatic glucose electrochemical sensor using a glassy carbon electrode modified by nanoporous copper sulfide doped with a high concentration of cobalt (Co-CuS-2). Glucose sensors are of great interest because one in ten adults is affected by diabetes, and the development of such nanomaterials is of particular interest because enzyme-free sensors have superior robustness in sensor performance.. Therefore, this research study is very suitable for this special issue. The experimental methods, results, and discussion of the results are appropriately scientifically sound.  Therefore, I believe this study is worthwhile for publications. However, as pointed out below, some of the background and explanation are not enough and the figures and errors should be corrected. Before publication, it is necessary to revise their manuscripts in the following points.

Comment

#1) minor error point (line 91): “ by X ray diffraction” is duplicated.

#2) The image in Figure 1 c-f is not high enough resolution to identify the element type (Cu, Co?) for each of the data). Please clarify the information in the figures or caption.

#3) The inset data in Figure 5b should be enlarged and clarified because the inset in Figure 5b is the main data in this section.

#4) I think the electrochemical response of sucrose does not appear to be small compared to the other interfering substances, although the effects of the interfering substances are seen. It would be better to show a quantitative graph comparing the glucose response with the effects of each interfering substance, as shown in ref. 25, so that the effects of each interfering substance can be compared.

#5) In order to evaluate it as a sensor, please add the experimental information about the reproducibility of its electrochemical response. For instance, how many sensor electrodes are produced and used for measurements? If the sensor number is not one, the sensor response values should have some error bars.  

#6) In order to provide a basic explanation of non-enzymatic electrochemical reactions, the following equation and its explanations should be added in the manuscript..

non-enzymatic electrochemical reaction equation

Cu (I) Cu (II) + e- …..(1)

Cu (II) + glucose Cu (I) + gluconolactone ……(2)

Author Response

1. Referee: 1 Comments: Authors employ CoCu sulfide derived from MOF for the detection of glucose. The work appears sound, however, there are some issues that need to be resolved before accepting the manuscript. They are listed below:

Comments: authors use abbreviations without explaining what they mean.

Response: We have explained the meaning of abbreviations in the revised manuscript.

Comments: analysis of sensing material has to be expanded. It would be nice to see the stoichiometry of metals and valance state of Co and Cu in the corresponding sulfide.

Response: The analysis of as-synthesized materials has to be expanded in this revised manuscript. We added EDS and XPS tests to analyze the stoichiometry of metals and valance state of Co and Cu in the corresponding sulfides. The element content of Co-CuS-1 and Co-CuS-2 samples were characterized by EDS, which indicated the Co/Cu ratios are close to the stoichiometric ratio of raw materials (Fig.4a). XPS technique was further used to analyze the surface valance state of Co and Cu in the corresponding sulfide. The high revolution spectra are shown in Figure 4b-d. The high revolution spectra of Cu in the three samples are similar, the binding energy of two peaks located at 931.4 and 951.3 eV is attributed to Cu 2p3/2 and Cu 2p1/2, respectively. The fitted peaks of 931.4 and 951.2 eV indicate the existence of Cu+, and the peaks at 932.4 and 953.1 eV corresponded to Cu2+. The high resolution Co 2p spectrum in Co-CuS-1 was fitted with two doublet peaks centered at 780.5 and 796.4 eV, which corresponds to Co3+. The peaks at 781.9 and 797.5 eV corresponded to Co2+. Figure 4d shows the S 2p spectra, the peaks centered at 161.2 and 162.4 eV for S 2p3/2 and S 2p1/2, respectively. For Co-CuS-2, the peak of 163.6 eV increased dramatically, which can be ascribed to metal-S bond at low coordination environment and contributed to increase the intrinsic conductivity.

Comments: how was LoD calculated? It seems very low in comparison with the art.

Response: In this manuscript, limit of detection (LOD) of the constructed sensors was calculated based on 3σ/s, where σ is the standard deviation of the background current and s is the slope of the calibration curves.

Comments: authors need to explain the redox processes during the voltammetric scan, especially reduction, and why such big hysteresis?

Response: In the revised manuscript, we explain the redox processes during the voltammetric scan. Redox peaks at 0.45/0.60 V (vs. Ag/AgCl) for Co-CuS-2 were observed on the CVs, which may be attributed to the redox reaction of Cu-CoS in the alkaline electrolyte

CuCo-S +OH- + H2O → CuSOH + CoSOH + e- (1)

CoSOH +OH- → CoSO + H2O + e- (2)

The big hysteresis in CV at high voltage is attributed to polarized and present oxygen evolution reaction.

Referee: 2

Comments: 1-When the authors cite Randles-Sevcik equation, please check the value number it would be 2.69 105, please also add the definition of each parameters and their units, as ex. D=7.6 10-6cm2s-1for FeIII(CN)63-, etc.

Response: Randles-Sevcik equation is not suitable for our constructed sensors, we deleted Randles-Sevcik equation in this revised manuscript.

2-in Fig. 5 b inset and in Figs. 6 d, e, f please add the error bars for all the data in the calibration plots and add in the text the replications of each mesurements.

Response: In the revised manuscript, we added the error bars for all the data in the calibration plots for Fig. 7 b inset and in Figs. 8 d, e, f in the revised manuscript.

Referee: 3 The author developed a highly sensitive electrochemical glucose sensor by using a nanoporous cobalt doped copper sulfide based electrode. In particular, they have succeeded in producing a highly sensitive non-enzymatic glucose electrochemical sensor using a glassy carbon electrode modified by nanoporous copper sulfide doped with a high concentration of cobalt (Co-CuS-2). Glucose sensors are of great interest because one in ten adults is affected by diabetes, and the development of such nanomaterials is of particular interest because enzyme-free sensors have superior robustness in sensor performance. Therefore, this research study is very suitable for this special issue. The experimental methods, results, and discussion of the results are appropriately scientifically sound. Therefore, I believe this study is worthwhile for publications. However, as pointed out below, some of the background and explanation are not enough and the figures and errors should be corrected. Before publication, it is necessary to revise their manuscripts in the following points.

Comments: 1) minor error point (line 91): “by X ray diffraction” is duplicated.

Response: The redundant words “by X ray diffraction” were deleted in the revised manuscript.

Comments: 2) The image in Figure 1 c-f is not high enough resolution to identify the element type (Cu, Co?) for each of the data). Please clarify the information in the figures or caption.

Response: In the revised manuscript, image in Figure 1 c-f was redrawn and shown in Figure 2 c-f.

Comments: 3) The inset data in Figure 5b should be enlarged and clarified because the inset in Figure 5b is the main data in this section.

Response: According to the suggestion of the reviewer, inset data in Figure 5b has been redrawn and enlarged in the revised manuscript as Figure 7b.

Comments: 4) I think the electrochemical response of sucrose does not appear to be small compared to the other interfering substances, although the effects of the interfering substances are seen. It would be better to show a quantitative graph comparing the glucose response with the effects of each interfering substance, as shown in ref. 25, so that the effects of each interfering substance can be compared.

Response: The glucose concentration of the human blood is actually 30-40 times higher than the interferents such as sucrose (Suc), ascorbic acid (AA), uric acid (UA), dopamine (DA), in this work, 20 μM interferents was successively injected into 0.1M NaOH containing 200 μM glucose. As shown in Fig. 9d, very weak responses after adding interferents for AA, UA, DA, Suc are observed, indicating that the sensor based on CuCo-S electrode has good selectivity.

Comments: 5) In order to evaluate it as a sensor, please add the experimental information about the reproducibility of its electrochemical response. For instance, how many sensor electrodes are produced and used for measurements? If the sensor number is not one, the sensor response values should have some error bars.

Response: In the revised manuscript, we added the experimental information about the reproducibility of its electrochemical response. The current responses of 0.2 mM glucose with the same Co-CuS-2 electrode were measured 13 times, and the relative standard deviation (RSD) is 4.19%. We added the error bars for all the data in the calibration plots for Fig. 7 b and in Figs. 8 d, e, f in the revised manuscript.

Comments: 6) In order to provide a basic explanation of non-enzymatic electrochemical reactions, the following equation and its explanations should be added in the manuscript.

non-enzymatic electrochemical reaction equation

Cu (I) ----Cu (II) + e- (1)

Cu (II) + glucose ----Cu (I) + gluconolactone (2)

Response: The possible oxidation mechanism of glucose may be described as follows

CuCo-S +OH- + H2O → CuSOH + CoSOH + e- (1)

CoSOH +OH- → CoSO + H2O + e- (2)

CuOSH + CoSO + glucose → CuS + CoSOH + glucolactone (3)

CuSOH and CoSO intermediator might be formed through electrooxidation in an alkaline medium. The formed CuSOH and CoSO adsorbed glucose molecules and subsequently converted to gluconolactone at alkaline conditions.

Reviewer 4 Report

In general this is a very well done and nice work with clear results and descriptions. I have only some few comments and suggestions.

line 147: A small suggestion: It would give a bit nicer/more serious look to show the mentioned Randles-Sevcik equation as a 'real equation' (using an equation editor) and using using squareroot signs instead of '1/2' exponents.

About the graphs presented a general remark. It could ease to identify the graphs by marking them in a better way.
For example the letter 'b' on the Figure 2b graph could be shown somehow differently than the axis title: in parenthesis '(b)' or in different colour or with some rectangular boundary line, etc.

line 361: Fig 2b the inset could be a bit bigger for better readibility. For example by moving the labels corresponding to the main curves to somewhere else (under the curve or above it next to the vertical axis)

line 185: 
Question/suggestion: 
Is it possible to strech somehow the linear region (to reach higher concentrations for sensing) It it is so what are the possibilities for that? 
I think the answer could be added to the text as a "future seeing/promising" note about these nice results.   

Author Response

In general this is a very well done and nice work with clear results and descriptions. I have only some few comments and suggestions.

Comments: line 147: A small suggestion: It would give a bit nicer/more serious look to show the mentioned Randles-Sevcik equation as a 'real equation' (using an equation editor) and using using square root signs instead of '1/2' exponents.

Response: Randles-Sevcik equation is not suitable for our constructed sensors, we deleted Randles-Sevcik equation in this revised manuscript.

Comments: About the graphs presented a general remark. It could ease to identify the graphs by marking them in a better way. For example the letter 'b' on the Figure 2b graph could be shown somehow differently than the axis title: in parenthesis '(b)' or in different colour or with some rectangular boundary line, etc.

Response: The graphs and Figures have been improved in the revised manuscript. In the revised manuscript, image in Figure 2b was redrawn and shown in Figure 3b.

Comments: line 361: Fig 2b the inset could be a bit bigger for better readibility. For example by moving the labels corresponding to the main curves to somewhere else (under the curve or above it next to the vertical axis)

Response: We have redrawn Fig 2b in the revised manuscript.
Comments: line 185: Question/suggestion:
Is it possible to strech somehow the linear region (to reach higher concentrations for sensing) It it is so what are the possibilities for that? I think the answer could be added to the text as a "future seeing/promising" note about these nice results.

Response: To extend the linear region to reach high concentrations for sensing glucose is very important. In general, to minimize the primary resistances and enhance the electrocatalytic sites of electrode is good strategy for improving electrochemical performances with wide linear region.